# Towards women-inclusive ecology: Representation, behavior, and perception of women at an international conference

**Anna Lupon** [1]*, **Pablo Rodríguez-Lozano**[2,3], **Mireia Bartrons**[4], **Alba Anadon-Rosell**[5,6], **Meritxell Batalla**[6], **Susana Bernal**[1], **Andrea G. Bravo**[7], **Pol Capdevila**[8,9], **Miguel Cañedo-Argüelles**[10], **Núria Catalán**[11], **Ana Genua-Olmedo**[12], **Cayetano Gutiérrez-Cánovas**[13], **Maria João Feio**[14], **Federica Lucati**[1,15,16], **Gabriela Onandia**[17,18], **Sílvia Poblador**[19], **Roser Rotchés-Ribalta** [6], **Anna Sala-Bubaré**[20], **María Mar Sánchez-Montoya** [21,22], **Marta Sebastián**[7], **Aitziber Zufiaurre**[6,23], **Ada Pastor**[24]

1 Integrative Freshwater Ecology Group, Centre d'Estudis Avançats de Blanes (CEAB-CSIC), Blanes, Spain, 2 Department of Geography, University of the Balearic Islands, Palma, Spain, 3 Department of Environmental Science, Policy, and Management, University of California at Berkeley, Berkeley, California, United States of America, 4 Aquatic Ecology Group, University of Vic—Central University of Catalonia (Uvic-UCC), Vic, Spain, 5 Landscape Ecology and Ecosystem Dynamics, Institute of Botany and Landscape Ecology, University of Greifswald, Greifswald, Germany, 6 CREAF, E08193 Bellaterra (Cerdanyola del Vallès), Catalonia, Spain, 7 Department of Marine Biology and Oceanography, Institut de Ciències del Mar (ICM-CSIC), Barcelona, Spain, 8 School of Biological Sciences, University of Bristol, Bristol, United Kingdom, 9 Department of Zoology, University of Oxford, Oxford, United Kingdom, 10 FEHM-Lab, Departament de Biologia Evolutiva, Ecologia i Ciències Ambientals, Institut de Recerca de l'Aigua (IdRA), Universitat de Barcelona, Barcelona, Spain, 11 Laboratoire des Sciences du Climat et de l'Environnement, LSCE, CNRS-UMR 8212, Gif Sur Yvette, France, 12 Centre for Environmental and Marine Studies (CESAM), Department of Biology, University of Aveiro, Aveiro, Portugal, 13 Doñana Biological Station (EBD-CSIC), Sevilla, Spain, 14 Department of Life Sciences, MARE-Marine and Environmental Sciences Centre, University of Coimbra, Coimbra, Portugal, 15 Centre for Ecology, Evolution and Environmental Changes (cE3c), University of Lisbon, Lisbon, Portugal, 16 Department of Political and Social Sciences, Universitat Pompeu Fabra (UPF), Barcelona, Spain, 17 Research Platform Data Analysis and Simulation, Leibniz Centre for Agricultural Landscape Research (ZALF), Müncheberg, Germany, 18 Berlin-Brandenburg Institute of Advanced Biodiversity Research (BBIB), Berlin, Germany, 19 Plants and Ecosystems (PLECO), Biology Department, University of Antwerp, Wilrijk, Belgium, 20 Faculty of Psychology, Education and Sports Sciences Blanquerna, Ramon Llull University, Barcelona, Spain, 21 Department of Ecology and Hydrology, International Excellence Campus for Higher Education and Research of the University of Murcia, Murcia, Spain, 22 Department of Biodiversity, Ecology, and Evolution, Faculty of Biological Sciences, Complutense University of Madrid, Madrid, Spain, 23 Área de Biodiversidad, Gestión Ambiental de Navarra-Nafarroako Ingurumen Kudeaketa (GAN-NIK), Pamplona-Iruña, Navarra, 24 Department of Biology, Aarhus University, Aarhus, Denmark

* alupon@ceab.csic.es

**Data Availability Statement:** The datasets that support the findings of this study and the codes

## Abstract

Conferences are ideal platforms for studying gender gaps in science because they are important cultural events that reflect barriers to women in academia. Here, we explored women's participation in ecology conferences by analyzing female representation, behavior, and personal experience at the 1st Meeting of the Iberian Society of Ecology (SIBECOL). The conference had 722 attendees, 576 contributions, and 27 scientific sessions. The gender of attendees and presenters was balanced (48/52% women/men), yet only 29% of the contributions had a woman as last author. Moreover, men presented most of the keynote talks (67%) and convened most of the sessions. Our results also showed that only 32% of

used for their analyses are available in Zenodo, with the identifier doi:10.5281/zenodo.5666619.

**Funding:** AL was supported by the Government of Catalonia and the the European Social Fund (ESF) through the program Beatriu de Pinós (BP-2018-00082). PR-L was supported by a Margalida Comas postdoctoral contract (PD/031/2018), funded by the Government of the Balearic Islands and the ESF. AA-R was supported by a Humboldt Research Fellowship. MB was supported by the Spanish Government through the project Alkaldia (PID2019-111137GB-C21). SB was supported by a Ramon y Cajal fellowship from the Spanish Government and AEI/FEDER UE (RYC-2017-22643). AGB was supported by a Marie Sklodowska-Curie (MSCA) Individual Fellowship (H2020-MSCA-IF-2016; project-749645). NC was funded by the European Union's Horizon 2020 research and innovation programme under the MSCA grant agreement No.839709. MJF was supported by the Portuguese Foundation of Science and Technology (FCT) through MARE strategic project (UIDB/04292/2020) and Norma Transitória. AGO was supported by the CESAM and FCT/MCTES (UIDP/50017/2020 + UIDB/50017/2020). CG-C was supported by the Spanish Government through a Juan de la Cierva – Incoporación contract (IJC2018-036642-I). FL had a doctoral grant funded by FCT (PD/BD/52598/2014). GO was supported by the German Federal Ministry of Education and Research within the Collaborative Project "Bridging in Biodiversity Science – BIBS" (01LC1501A-H). SP was supported by a postdoctoral fellowship MSCA Seal of Excellence of the Research Foundation – Flanders (12ZZS21N). The funders had no role in study design, data collection and analysis, decision to publish, or preparation of the manuscript.

**Competing interests:** The authors have declared that no competing interests exist.

the questions were asked by women, yet the number of questions raised by women increased when the speaker or the convener was a woman. Finally, the post-conference survey revealed that attendees had a good experience and did not perceive the event as a threatening context for women. Yet, differences in the responses between genders suggest that women tended to have a worse experience than their male counterparts. Although our results showed clear gender biases, most of the participants of the conference failed to detect it. Overall, we highlight the challenge of increasing women's scientific leadership, visibility and interaction in scientific conferences and we suggest several recommendations for creating inclusive meetings, thereby promoting equal opportunities for all participants.

## Introduction

Gender imbalances are pervasive in science, with women particularly underrepresented at senior academic positions [1]. While the demographic inertia from past policies may partially explain this trend [2, 3], gender bias is still evident in some of the key achievements that ground academic career progression, including the acquisition of prestigious grants or prizes [4], authorship positions in research articles [5], or invitations at conferences [6]. These gender imbalances, alarming by themselves, reduce the visibility of women researchers and might trigger vicious circles of gender bias that explain the persistence of *the glass ceiling* (*sensu* [7]). For example, low visibility leads to less awards and invitations to conferences that in turn lead to low reputation and career progression. This "invisibility" of women might also affect their well-being and motivation to pursue a scientific career and ultimately compromise the general quality of academia, which benefits from the integration of different perspectives [8]. Thereby, assessing the factors preventing the success of women in science is critical to enhance their success and an equitable future in scientific disciplines.

Academic conferences are crucial events for researchers' networking and exposure. These events are major scenarios to disseminate and learn about scientific advances, but offer also a public context where status and prestige may be displayed [9, 10]. Further, conferences bring opportunities for developing and fostering a broad network of collaborators [11], which is essential to boost academic success [12, 13]. However, not all researchers benefit equally from these events. Recent studies showed compelling evidence that women have reduced opportunities to participate in academic conferences compared to men. Discrimination against women has been reported in abstract selection [14], coauthor lists [15], and invitations to keynote talks [6, 16, 17]. Further, women speak less time than men when presenting their work [18] and ask fewer and shorter questions during "Questions and Answers" (Q&A) time [10, 19, 20]. One possible explanation for these trends is the demographic inertia, which states that gender biases in conferences mostly mirror imbalances in gender ratios among senior members because they are the ones that participate the most in these events. Hence, according to this theory, low participation of female researchers in conferences is a product of conditions that discouraged women to pursue an academic career in the past [21]. However, other less studied factors and non-recognized aspects such as those related with gender ideology [21] or implicit bias [10, 22] can also influence women's participation at conferences. The low participation and visibility of women at conferences has been previously associated with the so-called "chilly" academic environment [10, 23]. This phenomenon is built on implicit and explicit forms of sexism and incivility that signal to women that they do not belong in academia [24, 25]. Besides gender bias in participation, the "chilly" environment at conferences is often

related to attendees' attitudes, including the use of sexist expressions and stereotypical remarks, or the exclusion of women from intellectual discussions and social events [26]. Collectively, these "chilly" experiences may affect women's job satisfaction and lower their intentions to pursue a scientific career [23, 27]. Yet, studies analyzing conferences from a gender perspective are mostly restricted to gender representation and behavior, paying little attention to the perception of attendees (but see [23, 26]); while the few studies analyzing attendees' perception have not analyzed the participation of women. Hence, a joint analysis of gender representation, behavior, and perception in academic conferences is lacking.

Here, we explored the participation of women in ecology conferences by using the 1st Meeting of the Iberian Society of Ecology (SIBECOL, www.sibecol.org) as a testing ground. Specifically, we aimed to assess whether women and men were equally represented, participative and visible, and if not, to what extent these differences were due to demographic inertia or to other non-recognized aspects. To answer these objectives, we analyzed: (i) the representation of women in the attendance, speakers and organization panels, (ii) the audience behavior during Q&A times, and (iii) the perception of the attendees on women's participation and visibility, gender barriers, and conference environment (Fig 1). Such a multidimensional approach is critical to provide a comprehensive assessment of the situation of women in sciences, and to develop evidence-based policymaking promoting inclusive scientific conferences for female researchers.

## Methods

### Case study: The 1st Meeting of the Iberian Society of Ecology

We used as a case study the 1st Meeting of the Iberian Society of Ecology (SIBECOL; http://www.sibecol.org/), which was held in February 2019 in Barcelona, Spain. The conference gathered 722 researchers and held 576 contributions distributed in 27 scientific sessions (five general and 22 special sessions). All sessions were convened by teams composed from two to ten researchers, who proposed the topic, invited keynote speakers (max. one per session), selected the contributions, and chaired the session. For general sessions, these teams were composed by members of the conference scientific committee, while regular attendees (i.e., non-committee members) constituted the teams for special sessions. Among the contributions, there were 171 posters, 375 regular talks (12-min talk + 3-min Q&A time), 21 keynote talks (25-min talk + 5-min Q&A time), and nine plenary talks (50-min talk + 10-min Q&A time). SIBECOL society encouraged session conveners to be mindful in the selection of the talks among the contributions submitted to ensure gender equality and support young scientists.

### Attendance, authorships, and conveners

We examined gender distribution in attendance, presenters, authorships, and conveners by gathering information on the gender and career stage of all participants (Fig 1). Attendees self-reported their names, career stage, and the type of contribution they were presenting at registration, while the name of all coauthors and conveners was obtained from the book of abstracts. Then, the gender of each person was assigned as women or men based on their name (0.03% of the cases needed further information to determine the gender). The career stage, available for 549 participants, was classified in four categories: *pre-doctoral* (researchers not holding a PhD title, such as undergraduate, master and PhD students), *post-doctoral* (researchers not holding permanent positions that defended their PhD within the last eight years), *senior non-permanent* (researchers not holding permanent positions that defended their PhD more than eight years before the conference), and *senior permanent* (researchers holding permanent positions). Unfortunately, data related to other diversity axes (i.e., people

**Fig 1. Overview of the main objectives and the study design applied for analyzing gender biases in the 1ˢᵗ Iberian Ecological Society (SIBECOL) meeting.** We used a multidimensional approach including conference registration data (representation), observations during the event (behavior), and post-conference survey (perception). Icon source: www.flaticon.com.

belonging to underrepresented groups such as racial/ethnical minorities, LGBT+, disabilities or other gender identities) were not available. Therefore, it was not possible to analyze the participation of other genders and the intersection between gender and their belonging to other underrepresented groups and identities. Yet, it is worth noting that all attendees that responded to the post-conference anonymous online survey self-identified as either man or woman, despite the option "other gender" was also available.

We examined potential gender imbalances in attendance and presenters by comparing women and men's representation for all data pooled together, as well as for each contribution type and career stage separately. We also analyzed potential gender imbalances in authorships by analyzing the proportion of women and men signing as first and last author of all contributions. Based on previous studies in the field of ecology, we interpreted that the first author was the leader of the study, while the last author was considered to be the team leader or principal investigator of the project [28, 29]. Here, the first author was the presenter of the work in 97% of the cases according to the registration data. Further, we analyzed if there was associative gender sorting between first and last authors by applying a chi-squared test to Gaussian Generalized Linear Models (GLMs) [30]. For the models, we used the number of contributions as response variable and gender combinations of first-last authors (i.e. women-women, men-women, women-men, men-men) as fixed effect.

To investigate the gender distribution of conveners, we classified the sessions into three different categories: (i) sessions convened mostly by women (>60% of women conveners), (ii) sessions convened mostly by men (>60% of men conveners), and (iii) gender balanced sessions (40–60% of women/men conveners). We compared the number of sessions falling within each category by applying a chi-squared test to a GLM model [30]. We also quantified the number of sessions falling within each category that had a woman as keynote speaker, and

used a linear regression model to analyze whether the proportion of women in each conveners'
team influenced the proportion of women giving talks in their respective sessions.

## Audience and questioners during Q&A sessions

A team of 26 volunteers took notes on gender distribution and the questioning behavior of the
audience during the scientific sessions (Fig 1). Volunteers did not participate in the session
nor explain the study to third parts. This procedure allowed us to observe the behavior of the
attendees without conditioning it. To ensure that data from all observers was comparable, we
generated a standardized template with all questions and trained all volunteers prior the obser-
vations (S1 Appendix). For each talk, volunteers wrote down the number of attendees and
questioners by gender as well as the gender of the conveners and speaker (S1 Appendix). We
acknowledge that the gender recorded by the observers might not match with the gender iden-
tity of the person in some cases. In total, we collected data from 218 talks (56% of the total),
including seven plenaries, 12 keynote, and 199 regular talks. Poster contributions were not
analyzed. To account for differences among observers, 50% of the talks were evaluated by
more than one person (2–6 observers). For questions related to the number of attendees and
questioners, we averaged the values obtained from all observes (inter-observers' variability
was < 5%). For gender-related questions (i.e., those whose answer was "man" or "woman"),
the most frequent answer was computed. Yet, discrepancies in gender-related questions only
occurred on five occasions (2.3% of the talks).

For each talk type, we used t-tests to analyze (i) if the gender distribution of the audience
was similar to that of the conference registration and (ii) whether the number of attendees dif-
fered between female and male speakers. We also examined whether the speaker's gender
influenced the gender distribution of attendees by comparing two Linear Mixed-effect Models
(LMM), one including and one excluding the gender of the speaker as independent variable.
We considered that the speaker's gender was related to the gender of the attendees if the differ-
ence in the Akaike Information Criterion (AIC) between models were higher than 2 (*sensu*
[31]). To account for the lack of independence between sessions (i.e., participants could move
from one room to another between talks), we used session as a random effect.

For all talks pooled together, we analyzed the number of questions raised by gender and
how it varied depending on the gender of the speaker and the gender of the convener moderat-
ing the session. First, we analyzed the representation of women by plotting the proportion of
questions raised by women compared to the proportion of women in the audience following
[32]. A relationship of 1:1 indicates that women asked questions proportionately to their repre-
sentation in the audience, while deviations from that line indicate that women asked more
(odds ratio > 1) or less (odds ratio < 1) questions than expected. Moreover, a slope different
than 1 indicates that the questioning behavior of women varies with the proportion of women
in the audience. Then, we used chi-squared tests to analyze whether the number of questions
or the gender of questioners differed between talks given by women and men. To avoid statisti-
cal biases, we standardized the number of questions raised by each gender by the number of
attendees of that gender (i.e., number of questions made by women / number of women in the
audience). The same analysis was done to test whether the questioning behavior of the audi-
ence differed between talks convened by women or men.

## Attendees' perception of the conference

We examined the attendees' perception of the conference with a post-conference anonymous
online survey (Fig 1; S2 Appendix). During the weeks following the conference, we invited all
attendees to participate. Two reminders were sent two and three weeks after the conference,

and the survey was closed a month after the conference. The survey took about 10 minutes to complete and was answered by 32% of attendees (n = 232). This response rate was larger than those reported in other gender studies, for both women and men (typically from 10 to 20%; [33]). The survey included a written informed consent and closed questions about their participation and perception of the conference, divided in four main sections: (i) questions to all attendees related to their participation and perception during Q&A time, (ii) questions only for speakers regarding their perception of the Q&A time proceeding their talks, (iii) questions to all attendees about their satisfaction and perception of the conference, oriented to detect a potentially "chilly" environment, and (iv) socio-demographic inquiries. Questions regarding Q&A time were based on [32] and aimed to analyze attendees' perception, as well as their satisfaction as speakers. Questions related to "chilly" environment were adapted from previous literature [23, 26, 34] and evaluated if respondents (i) heard any gender stereotypical remark, (ii) experienced uncivil behaviors, (iii) were intellectually and socially satisfied, and (iv) suffered from the impostor syndrome (i.e., felt like a fraud; [35]). Most questions in the survey were closed-ended, i.e., we used five-point Likert scale questions to measure the level of agreement or frequency related to different statements. We acknowledge that the survey results might be skewed towards those attendees who were more aware of gender issues. However, it is impossible to test for the occurrence of this phenomenon, and if it occurred, it was applicable for both women and men. Hence, we consider that it did not affect the results.

For each question, we fitted GLMs using Gaussian error distributions for Likert scale questions and Binomial error distributions for binary response variables. Each model included gender and career stage as predictors. Final models did not consider the interaction between gender and career stage because it was not significant in any initial model. Then, we used chi-squared tests to determine whether respondents' gender or career stage influenced their answers.

We ran all statistical tests in R 3.6.1 [36]. For all analyses, differences were considered significant when $p$-value $< 0.05$. In all cases, model residuals were visually inspected to verify linear model assumptions of normality and homoscedasticity. The code and data used in this study can be accessed in https://github.com/SIBECOL-Diversity-Inclusion/SIBECOL_gender_inequality. This study and all the analyses therein was approved by the Committee for Protection of Human Subjects of the University of California at Berkeley that serves as an Institutional Review Board (IRB) (protocol number #2019-03-12006).

## Results

### Attendance, authorships, and conveners

The conference was gender balanced in terms of attendance (48% women vs 52% men) and presenters (50% women vs 50% men). The proportion of female presenters decreased with seniority, accounting for the 60% of pre-doctoral presenters, but only for the 40% of senior permanent presenters (S1 Table in S3 Appendix). Further, the proportion of female presenters differed among contribution types: almost half of the presenters were women for poster (55%), regular (49%) and plenary (44%) contributions, while only the 33% of keynote talks were presented by women (S1 Table in S3 Appendix).

Women and men were first authors of the same number of contributions (50% women vs 50% men), while only 29% of last authors were women. When considering both the first and the last authorship, there were gender imbalances ($\chi^2$ test, $p < 0.001$). Specifically, 40% of the contributions had men as first and last authors, while only 17% had women as first and last authors. The combination with lower representation (11%) was men as first author and women as last author (Fig 2A).

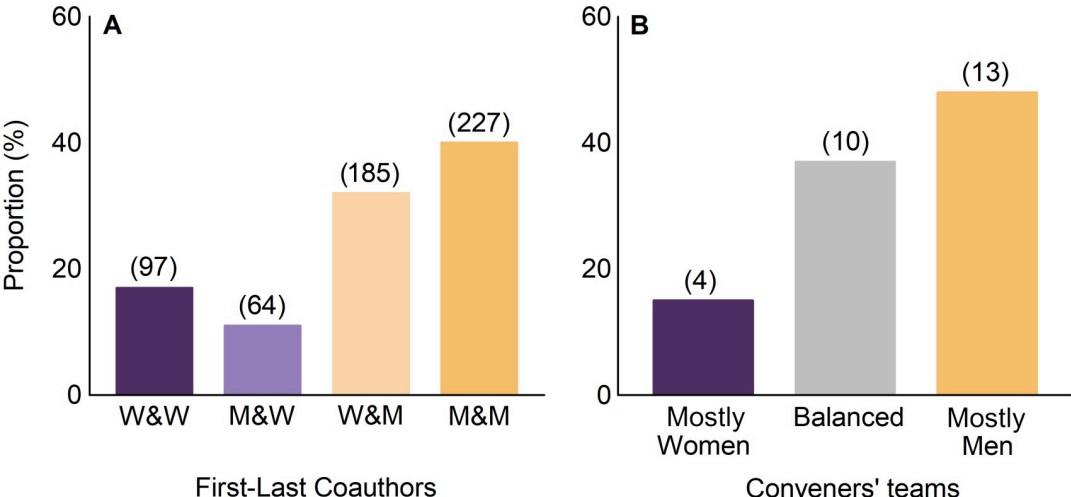

**Fig 2. Gender distribution of coauthors and conveners.** (**A**) Gender combination of researchers that signed as first and last coauthors in the contributions of the 1st SIBECOL Meeting (gender first author & gender last author; W = woman and M = man). We interpreted that the first author was the leader of the presented work, while the last author was the principal investigator of the research group or project. (**B**) Proportion of sessions whose conveners were mostly women (purple bar; > 60% of the conveners were women), equally distributed (grey bar; 40–60% of conveners were women), and mostly men (yellow bar; > 60% of conveners were men). The number of contributions (panel A) or sessions (panel B) falling within each category is shown in parenthesis.

Women represented 31% and 30% of the conference organizing and scientific committees, respectively. Further, from all sessions, 37% were convened by a balanced proportion of women and men (between 40 and 60% women/men), 48% were mostly convened by men, and only 15% were convened mostly by women (Fig 2B). All the sessions with mostly female conveners were special sessions, whereas general sessions had either an equal or higher number of male conveners. There was no relationship between the gender of conveners and presenters (linear regression, $R^2 = 0.04$, $p = 0.314$). However, 66% of the sessions convened mostly by women had a woman as keynote speaker, while this percentage was 33% and 22% for those sessions that had equal or higher number of men conveners, respectively.

### Audience and questions during talks

For the analyzed talks, there was a similar proportion of women and men in the audience (40–60%). This proportion did not differ from the proportion of women and men registered at the conference (for all talk types: t-test, $p > 0.070$). We did not find any relation between the gender of the speaker and the gender of the attendees (AIC diff. < 2). However, the number of attendees differed between female and male speakers (for all talk types: t-test, $p < 0.020$). In particular, the number of attendees was 12%, 7% and 18% higher when the speaker was a man than when it was a woman for the regular, keynote, and plenary talks, respectively.

Among the 218 analyzed talks, 47 (22%) did not get any question. The gender of the speaker or convener had no impact on whether the talk had questions (in both cases: $\chi^2$ test, $p > 0.3$). For the 171 presentations that received questions, the average number of questions per talk was similar for both female and male speakers (1–2 questions; t-test, $p = 0.491$) (S2 Table in S3 Appendix). However, on average, women asked fewer questions than expected given their representation in the audience (odds ratio = 0.7; Fig 3) and this phenomenon persisted regardless the proportion of women in the audience (slope test, $p = 0.571$). Moreover, the number of questions raised by women varied depending on the gender of the speaker and convener.

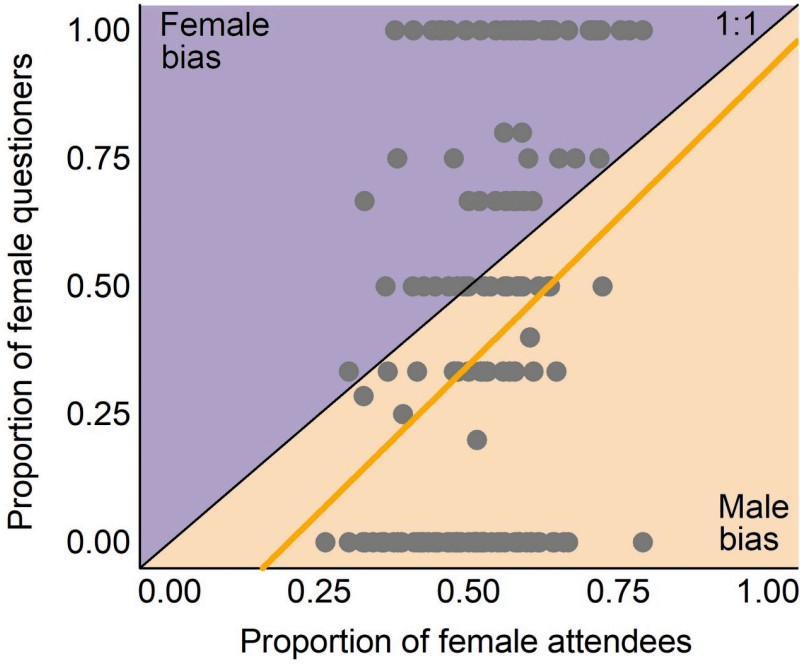

**Fig 3. Proportion of questions asked by women plotted against the proportion of women attendees during the analyzed Q&A sessions.** The black line shows the theoretical proportional relationship between the two parameters. The yellow line shows the real proportional relationship based on a linear regression of the data (mean odds ratio: 0.7).

Female speakers received a similar number of questions from women than men (t-test, $p = 0.079$), while male speakers received significantly more questions from men than from women (t-test, $p < 0.001$) (Fig 4A). Likewise, women and men asked a similar number of questions when the convener was a woman (t-test, $p = 0.096$), while men asked more questions than women when the convener was a man (t-test, $p < 0.001$) (Fig 4B).

From all the talks that received questions, a woman asked the first question in 37% of the cases (63 out of 171 talks). There were no differences in the gender of the person asking the first question when the speaker was a woman ($\chi^2$ test, $p = 0.070$), while the first question was asked by a man in 67% of the cases when the speaker was a man ($\chi^2$ test, $p < 0.001$). The gender of the convener was not related with the gender of the person asking the first question ($\chi^2$ test, $p = 0.109$). No relationship was found between the gender of the person asking the first and subsequent questions ($\chi^2$ test, $p = 0.138$).

## Attendees' perception of the conference

Within the survey respondents (n = 232), 60% were women, 38% were men, and 2% did not answer this question (no respondent marked the "other gender" response option). Further, 31% of the respondents were pre-doctoral, 37% were post-doctoral, 11% were senior non-permanent, and 19% were senior permanent researchers. Most respondents had Spanish nationality (83%) and were living in Spain when the conference was held (75%). Respondents mostly worked at universities (58%) or research institutions (34%). Only a few respondents reported belonging to minority groups: LGBT+ (4.3%), race/ethnic minorities (1.3%), and people with disabilities (0.4%).

Related to how attendees perceived the gender distribution of questioning, 32% of respondents perceived that men asked more questions than women during Q&A time. This

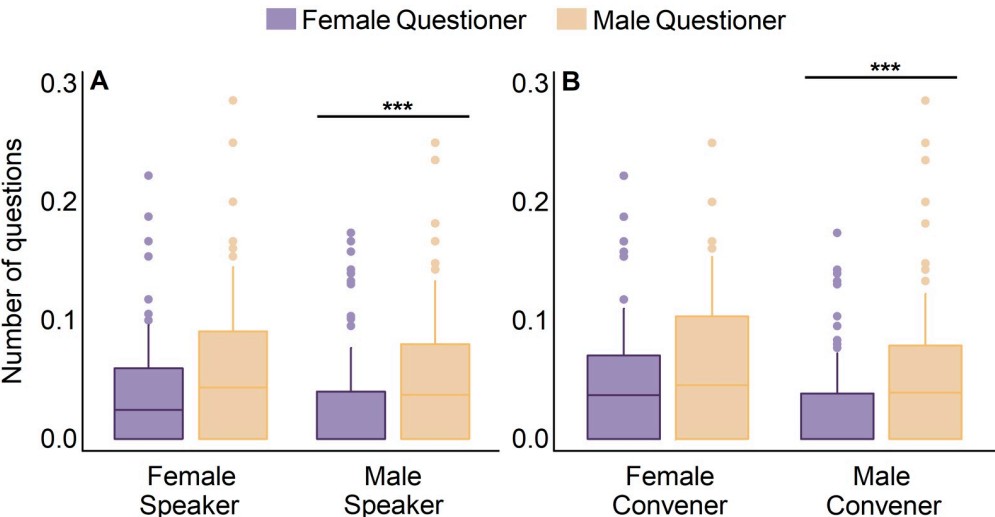

**Fig 4. Number of questions per talk raised by women (purple) and men (yellow) considering the gender of (A) the speaker and (B) the convener.** In all cases, the number of questions raised by each gender is standardized by the number of attendees of that gender (i.e., number of questions made by women / number of women in the audience). Asterisks indicate significant differences between the number of questions asked by women and men (t-test, $p < 0.05$).

perception was significantly higher among women (39%) than among men (21%) ($\chi^2$ test, $p < 0.001$) (Fig 5A). In this vein, a higher proportion of men than women reported "to have always asked questions when they wanted to" (38% vs 25%; $\chi^2$ test, $p = 0.009$). This difference between male and female attendees was the highest for senior non-permanent researchers and the lowest for senior permanent researchers (Fig 5B).

From the respondents that contributed with a talk, 88% agreed or strongly agreed that the convener of the session was not gender biased when choosing the questioners. Further, their experience as speakers was positive regardless of their gender or career stage (for all questions related to speakers' satisfaction: $\chi^2$ test, $p > 0.05$; see Table 1).

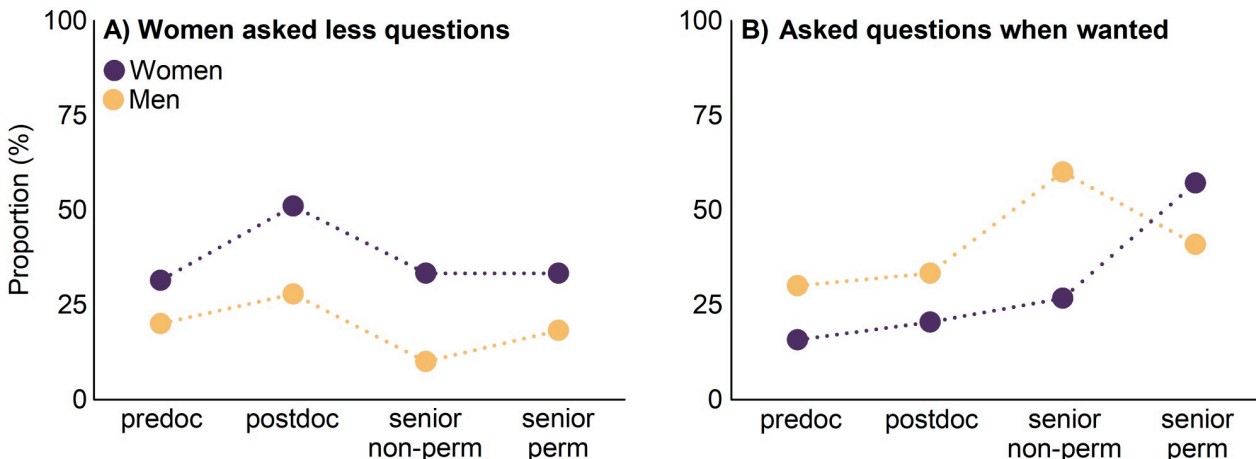

**Fig 5. Proportion of participants that answered in the post-conference survey (A) "women asked less questions than men during oral communications" and (B) "I always asked questions when I wanted to".** The proportion is shown for each gender separately (e.g., women that responded to a specific answer/total number of women that participated in the survey). Abbreviations: master students and predoctoral researchers (predoc), postdoctoral researchers (postdoc), senior researchers with non-permanent positions (senior non-perm), senior researchers with permanent positions (senior perm).

**Table 1. Summary of the responses to the five-point Likert scale questions included in the post-conference survey.**

| Question | Women | Men |
|---|---|---|
| *Questions to speakers about the Q&A time after their talk* | | |
| Q1. The convener was gender biased when choosing questioners | 1.51 ± 0.79 | 1.43 ± 0.81 |
| Q2. The questions were constructive | 4.16 ± 0.74 | 4.24 ± 0.71 |
| Q3. The questions were formulated politely | 4.57 ± 0.52 | 4.47 ± 0.54 |
| Q4. I felt satisfied with my answers | 4.04 ± 0.83 | 3.92 ± 0.80 |
| *Questions to all attendees about the conference* | | |
| Q5. I suffered the impostor syndrome, i.e., I felt like a fraud | **2.73 ± 1.20** | **2.22 ± 1.18** |
| Q6. Others came to me to discuss intellectual ideas | 3.52 ± 1.04 | 3.78 ± 0.79 |
| Q7. I am satisfied with the level of intellectual stimulation | 4.06 ± 0.80 | 4.16 ± 0.78 |
| Q8. I am satisfied with the amount of social interactions with others | 3.66 ± 0.95 | 3.90 ± 0.88 |
| Q9. I think the conference supported work-life balance | 3.40 ± 0.97 | 3.53 ± 0.91 |
| Q10. In general, I liked attending the conference | 4.28 ± 0.69 | 4.37 ± 0.74 |
| Q11. How often did you hear gender stereotypical remarks? | **1.61 ± 0.78** | **1.40 ± 0.63** |
| Q12. How often did you feel excluded from social activities? | 2.03 ± 0.91 | 1.93 ± 0.94 |
| Q13. How often did someone put you down or was mean to you? | 1.31 ± 0.61 | 1.22 ± 0.58 |
| Q14. How often did someone pay little attention to your statement? | 2.05 ± 0.98 | 1.92 ± 0.94 |

Questions 1–10 measured the level of agreement with a statement (where 1 was "strongly disagree" and 5 was "strongly agree"), while questions 11–14 measured the frequency (where 1 was "never" and 5 was "very frequently"). Values in bold indicate significant statistical differences between women and men responses.

Regarding the conference environment, most respondents were satisfied with the conference regardless of their gender (in all satisfaction related questions: $\chi^2$ test, $p > 0.05$; Table 1). However, the degree of satisfaction differed across career stages (in most satisfaction related questions: $\chi^2$ test, $p < 0.05$). Specifically, and regardless of their gender, senior permanent researchers reported to have experienced a greater intellectual stimulation during the conference and to have felt less excluded from social activities, compared to pre-doctoral and post-doctoral researchers. Further, 60% of the respondents never heard any gender stereotypical remark during the conference and none of the respondents reported to have heard gender stereotypical remarks very frequently. Despite this general positive perception, women self-reported to have heard more gender stereotypical remarks than men ($\chi^2$ test, $p = 0.04$; Table 1). Particularly, 43% of women reported to have heard some stereotypical remarks, while only 34% of the men did. Moreover, more female than male respondents self-reported to have heard stereotypical remarks occasionally or frequently (17% vs. 6%). Finally, 28% of the respondents agreed or strongly agreed to have suffered the impostor syndrome during the conference, and this percentage was higher for women than for men (34% vs 19%; $\chi^2$ test, $p = 0.008$; Table 1) and decreased with career stage ($\chi^2$ test, $p = 0.030$).

## Discussion

This study aimed to analyze the role and visibility of women at a major conference of ecology in the Iberian Peninsula from a multidimensional perspective: representation, behavior, and personal perception (Fig 1). While we found evidence that gender imbalances persist in terms of representation and in-conference behavior, most attendees, either women or men, did not perceive this imbalance. These findings highlight persistent hurdles on the path to achieve gender equity in scientific events, and thus stress the importance of designing and implementing further actions for advancing towards complete inclusive scenarios.

## Women's representation: Beyond the demographic inertia problem

Compared to other conferences (e.g. [16, 37]), the 1st SIBECOL Meeting was gender balanced in terms of attendance, presenters, first-authors, and plenary speakers. However, and despite of the provided recommendations on gender equality by the organizing committee, and the fact that the conference was the kickoff of the SIBECOL society (thus preventing past inertias), women were underrepresented in various scientific domains related to the conference. Women only accounted for 33% of keynote speakers and 29% of last authorships. Further, women were a minority (< 40%) in both organizing and scientific committees, as well as in 48% of conveners' teams. These results are consistent with previous studies showing that, even though the field of ecology is generally female-dominated at the student level, women are still under-represented in senior and prestigious academic positions [6, 17, 38, 39].

One possible explanation for the reduced number of women as keynote speakers or conveners is the so-called demographic inertia, as senior researchers are generally the ones that participate the most in these events [2, 20]. However, our results partially challenge this hypothesis because women accounted for > 40% of senior researchers registered in the conference. Hence, we argue that differences in gender representation were likely related to other factors, including implicit bias (i.e., devaluation of women's work; [40]) or the tendency of women to decline invitations and leadership roles because of a higher share of domestic tasks, and parental or elderly care [41]. Additionally, associative gender sorting can contribute to maintaining gender gaps at conferences [16]. In this line, we found that 88% of the sessions with a majority of male conveners had a man as keynote speaker, while the keynote speaker was a woman in 66% of the sessions with a majority of female conveners. Previous research has also observed non-random associations of genders in all fields of science and proposed several reasons for it, such as that women and men have different research interests [42]; that women more often consider gender issues and make conscious efforts to find female speakers or mentees [16]; or that people are more likely to collaborate with someone "similar" to oneself [43]. Regardless of the cause, our findings show that women were underrepresented in both authorships and conveners' teams, which harms the visibility of female researchers and has potential repercussions for young women's motivation and withdrawal in the field of ecology.

## Session environment and behavioral dynamics

Observations of in-conference behaviors identified several attitudes and practices that led to gender bias. We found that, regardless of the contribution type, the attendance was ~10% lower during talks performed by women. This difference in attendance might be explained by the lower recognition and visibility of women in sciences [44] or by the existence of an implicit bias against women that influences the evaluation of scientist's work [45]. There is evidence that unconscious devaluation of women is widespread in science [40, 46] and affects personal choices such as hiring and decision making [4], building a network or collaborations [38] or, in this case, attending a particular talk. Our results further indicate that judgments made by women and men researchers are equally influenced by this implicit bias, as the gender distribution of attendees did not differ between female and male speakers. These results agree with the idea that women are still considered less competent in science than men by both female and male researchers [45].

Another concerning result was the lower participation of women compared to men during Q&A time. Different participation of men and women on "stage-time" can have various potential causes, including differences in gender demographics, ideology, self-esteem, and behavior [10, 47, 48]. For instance, women are more likely to raise their hand if they do not have to

compete to be called on; that is, if Q&A time is longer or there are only few raised hands [49]. However, current "stage-time" dynamics favor competition and are usually dominated by those attendees that have more authority or have a close relationship with the convener, which are generally men [22, 32, 50]. The solution to this imbalance may not be urging women to behave more aggressively during Q&A times, but rather to rethink the structure and dynamics of Q&A sessions to foster inclusive participation. Accordingly, women self-reported asking questions whenever they wanted less frequently than men at most career stages. The only exception was for senior permanent researchers, where more women than men self-reported to have asked questions whenever they wanted, which might suggest an increased feeling of community belonging of those women that "survived" the leaky pipeline [51].

Interestingly, the audience behavior changed depending on the gender of conveners and speakers. We observed that men mostly asked questions to male speakers, while women, as both speakers and conveners, promoted gender balance in questioners. Further, the first questioner was generally a man when the speaker was a man, while a similar number of women and men asked the first question when the speaker was a woman; which agrees with observations in other academic fields [10, 32]. These findings suggest that, even when the structure of the Q&A time does not suit your personality or ideology, asking a question is easier and less intimidating when there is a sense of familiarity within the session [47, 52]. Ultimately, this finding agrees with the "stereotype inoculation model" [53], suggesting that female speakers and conveners serve as role models that increased the sense of belonging and mitigated stereotype threats, such as that women are less capable of succeeding in science.

## Attendees perception: Taking the temperature of the conference

Results from the post-conference survey show that, in general terms, attendees had a good experience during the 1<sup>st</sup> SIBECOL Meeting and did not perceive the event as a threatening context for women. However, subtle distinct responses between genders also indicate that women tended to have a worse experience than their male counterparts, at least among the survey respondents. We noted that women self-reported hearing stereotypical remarks more often than men, and more women than men noted gender disparities during the Q&A time. Moreover, even if not significant, women tended to self-report less satisfaction and more exclusion than men. Collectively, these findings indicate that women are more aware of and sensitive to gender issues compared to men, probably because women are more likely to be the target of sexist attitudes [23, 54]. More importantly, the abovementioned results suggest that the conference might have been a potentially "chilly environment" for women (*sensu* [23]). We argue that this "chilly environment" did not happen through overt displays of sexism or intolerance, but rather through the accumulation of subtle behaviors (e.g., keynote speaker selection, audience, Q&A dynamics). Accordingly, previous studies have reported that low female representation, competitive Q&A dynamics, and the indirect exclusion of women from social events such as after-conference activities that compromise the participation of women with family responsibilities, give subtle cues of non-belonging to women that might contribute to their intentions to quit science [10, 23, 50].

Finally, our results show that women self-reported to experience the impostor syndrome more often than men at every career stage. The causes of the impostor syndrome are complex, and many times derive from individual, social, and cultural stereotypes that go beyond conferences and academia [55]. The impostor syndrome may affect one's self-esteem and ultimately influence their behaviors and practices [55, 56]. Therefore, we cannot rule out that impostor syndrome thoughts may have induced the behavior of some women during the conference. In turn, it is also possible that the gender imbalances detected in this conference contribute to

increase women's impostor syndrome thoughts. Low visibility and recognition of women's work (i.e., differences in keynote speakers, conveners, or attendance) might lead women to partially believe that their research is not good enough to attract other researchers' interest, or that senior positions and recognition within academia is exclusively for men.

## Moving towards women's inclusive scientific conferences

Our research highlights gender disparities in the levels of participation in an ecology conference, thus stressing the importance of advancing towards more women-inclusive conferences, even in First World countries with good perceptions of women as scientists [57, 58]. We strongly encourage organizers to develop and follow guidelines for inclusive scientific meetings like those previously reported in [59, 60], with special focus on:

*Increasing the involvement and visibility of women in conferences*—In line with previous studies [16, 37, 61], our results point out that involving women as conveners or keynote speakers is an effective and practical action for achieving gender balance at scientific conferences because women's participation increased, and even equaled men's participation, when speakers and conveners were women. To date, most efforts in ecology conferences, including the 1st SIBECOL Meeting, have been focused on achieving gender parity in plenary talks; however, including women at different levels of conference organization (from the scientific committee to conveners) could enhance the participation of female attendees and reinforce their feeling of belonging. With that aim, clear open guidelines and criteria, including rules of gender quotas, open calls for participation, promotion of female scientist' directories or an open explanation for the reasons of choosing a particular keynote speaker, should be designed and provided to conference organizers.

*Speaking out about gender imbalances*—One of our most concerning results was to observe that most survey respondents did not notice disparities in gender participation during Q&A time. In this line, we recommend to include some activities that promote the public discussion of gender related issues during conferences, such as plenary talks, round tables and workshops; or just add an initial statement at the start of each session to raise awareness on inclusive practices [59]. These activities provide the opportunity to unmask prevalent subtle disparities, and thus, could improve diversity justice in future conferences [47]. For instance, when the conferences of the Iberian Society of Limnology (AIL; www.limnetica. es) started incorporating a special session on Gender & Science in 2014, the number of women as plenary speakers increased in the following meetings [62]. Therefore, including at least one activity for raising awareness in inclusiveness during conferences should be a top priority for scientific societies.

*Redesigning conferences to foster equity and inclusion*—Our study suggests that further work is needed to create comfortable environments to promote the full participation of women during conferences. The first step towards this direction is to carefully choose the event's location, materials, and policies to ensure positive, pro-active attitudes towards diversity and inclusion [63]. In addition, we should rethink and redesign sessions' dynamics to include all ideologies and capabilities. Therefore, we urge the consideration of alternative settings of discussion in academic settings, which might include small group discussions, use of technology/social media as support for Q&A session, or speed-dating activities to promote discussion among participants within a more relaxed environment. Also, some recommendations to the conveners on how to deal with timekeeping issues and reformulate misplaced questions might help to establish a more respectful and mindful environment

during the sessions. Those alternative settings should be carefully evaluated and designed to meet equity participation targets.

*Involving ecological societies as allies*—Ecological societies should play a pivotal role for inclusivity, as they represent academic culture and ethics benchmarks of their fields [64]. In recent years, several societies have stepped forward towards this direction by initiating discussions about how to promote diversity or implementing initiatives to directly or indirectly promote inclusiveness. Examples of these initiatives include adopting a Code of Conduct, such as the ones of the European Geosciences Union (EGU https://www.egu.eu/about/code-of-conduct/) or the Society for Freshwater Science (SFS) Meeting (https://sfsannualmeeting.org/) support for child care, or spaces equipped to accommodate the needs of nursing mothers [65]. In addition, the organization of training courses on gender bias or gender inclusive language [66], creating mentor-mentees programs [63] or fostering collaboration among early-career or women researchers [67] can contribute to avoid gender imbalances.

Finally, we should highlight that women are not the only underrepresented group that has been historically excluded in academia. Previous studies have shown that researchers belonging to minority groups (e.g., non-binary, LGBT+ or color people) have reduced participation and visibility at conferences, especially if they are women [68–70]. In the post-survey of the 1st SIBECOL Meeting, none of the respondents self-reported to have a gender identity different to male or female, and only a small percentage (~6%) of attendees reported to belong to other minorities, which prevent us to include these aspects in our analysis. The low attendance of researchers belonging to these underrepresented groups is alarming by itself, and stresses the existence of systemic inequalities that hinder academic career pursuing. If the aim of ecology conferences is to advance knowledge for the betterment of the world as a whole, academic societies should seek to demolish existing structural barriers for these minority groups and ensure environments that are safe and comfortable for everyone, regardless of their gender identity, gender expression, sexual orientation, ethnicity, physical or mental difference, religion, or national origin.

## Supporting information

**S1 Appendix. Template used for all the observers to evaluate the attendance and participation during sessions.**
(PDF)

**S2 Appendix. Complete form of the post-conference on-line survey.** Note that we only present selected results here.
(PDF)

**S3 Appendix. Summary of the collected data.** It includes S1 and S2 Tables.
(PDF)

## Acknowledgments

We thank the support of the Iberian Society of Ecology (SIBECOL), in particular the president, Cèlia Marrasé, and the conference organizing committee. We are very grateful to the team of volunteers that helped collecting observation data during the conference (alphabetically): Mireia Banqué, Laura Blanquer, Míriam Colls, Veronica Granados Perez, Margarita Menéndez, Laura Roquer, Jordi René-Mor, Clara Ruiz-González, and Cristina Romera-Castillo. We also

thank Dolly Kothawala and Anne M. Kellerman for English revisions. One anonymous reviewer and Sandra Hille helped to improve an early version of this manuscript.

## Author Contributions

**Conceptualization:** Anna Lupon, Pablo Rodríguez-Lozano, Anna Sala-Bubaré, Ada Pastor.

**Data curation:** Anna Lupon, Pablo Rodríguez-Lozano, Mireia Bartrons, Meritxell Batalla, Andrea G. Bravo, Pol Capdevila, Cayetano Gutiérrez-Cánovas.

**Formal analysis:** Anna Lupon, Pablo Rodríguez-Lozano, Mireia Bartrons, Alba Anadon-Rosell, Meritxell Batalla, Susana Bernal, Andrea G. Bravo, Pol Capdevila, Miguel Cañedo-Argüelles, Núria Catalán, Ana Genua-Olmedo, Cayetano Gutiérrez-Cánovas, Maria João Feio, Federica Lucati, Gabriela Onandia, Sílvia Poblador, Roser Rotchés-Ribalta, Anna Sala-Bubaré, María Mar Sánchez-Montoya, Marta Sebastián, Aitziber Zufiaurre.

**Investigation:** Anna Lupon, Pablo Rodríguez-Lozano, Alba Anadon-Rosell, Meritxell Batalla, Susana Bernal, Andrea G. Bravo, Miguel Cañedo-Argüelles, Núria Catalán, Ana Genua-Olmedo, Maria João Feio, Federica Lucati, Gabriela Onandia, Sílvia Poblador, Roser Rotchés-Ribalta, María Mar Sánchez-Montoya, Marta Sebastián, Aitziber Zufiaurre, Ada Pastor.

**Methodology:** Anna Lupon, Pablo Rodríguez-Lozano, Mireia Bartrons, Pol Capdevila, Cayetano Gutiérrez-Cánovas, Anna Sala-Bubaré, Ada Pastor.

**Project administration:** Anna Lupon, Pablo Rodríguez-Lozano, Mireia Bartrons, Ada Pastor.

**Validation:** Anna Lupon, Pablo Rodríguez-Lozano, Mireia Bartrons, Meritxell Batalla, Pol Capdevila, Cayetano Gutiérrez-Cánovas.

**Visualization:** Anna Lupon, Pol Capdevila.

**Writing – original draft:** Anna Lupon, Pablo Rodríguez-Lozano, Ada Pastor.

**Writing – review & editing:** Mireia Bartrons, Alba Anadon-Rosell, Meritxell Batalla, Susana Bernal, Andrea G. Bravo, Pol Capdevila, Miguel Cañedo-Argüelles, Núria Catalán, Ana Genua-Olmedo, Cayetano Gutiérrez-Cánovas, Maria João Feio, Federica Lucati, Gabriela Onandia, Sílvia Poblador, Roser Rotchés-Ribalta, Anna Sala-Bubaré, María Mar Sánchez-Montoya, Marta Sebastián, Aitziber Zufiaurre.

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
