## [Decision Letter · Decision Letter 0]

15 Jul 2021

PONE-D-21-16326

Towards women-inclusive ecology: Representation, behavior, and perception of women at an international conference

PLOS ONE

Dear Dr. Lupon,

Thank you for submitting your manuscript to PLOS ONE. After careful consideration, we feel that it has merit but does not fully meet PLOS ONE’s publication criteria as it currently stands. Therefore, we invite you to submit a revised version of the manuscript that addresses the points raised during the review process.

Both reviewers pointed out the novelty of the submitted manuscript and necessity of such studies to better understand how women interact within the academic environment. As you will see, both reviewers pointed out several issues that should be improved. Especially, Reviewer 1 asks to better explain how representative is the outcome of the study based on the data. Reviewer 2 highlights that parts of the manuscripts especially abstract, introduction and discussion should be clearer and more concise.

We look forward to receiving your revised manuscript.

Kind regards,

Ute Risse-Buhl, Ph.D.

Academic Editor

PLOS ONE

Journal Requirements:

We note that one or more of the authors are employed by a commercial company: AECOM-URS.

2.1. Please provide an amended Funding Statement declaring this commercial affiliation, as well as a statement regarding the Role of Funders in your study. If the funding organization did not play a role in the study design, data collection and analysis, decision to publish, or preparation of the manuscript and only provided financial support in the form of authors' salaries and/or research materials, please review your statements relating to the author contributions, and ensure you have specifically and accurately indicated the role(s) that these authors had in your study. You can update author roles in the Author Contributions section of the online submission form.

2.2. Please also provide an updated Competing Interests Statement declaring this commercial affiliation along with any other relevant declarations relating to employment, consultancy, patents, products in development, or marketed products, etc.  

Reviewers' comments:

Reviewer's Responses to Questions

**Comments to the Author**

1. Is the manuscript technically sound, and do the data support the conclusions?

Reviewer #1: Partly

Reviewer #2: Yes

2. Has the statistical analysis been performed appropriately and rigorously? 

Reviewer #1: Yes

Reviewer #2: Yes

3. Have the authors made all data underlying the findings in their manuscript fully available?

Reviewer #1: Yes

Reviewer #2: Yes

4. Is the manuscript presented in an intelligible fashion and written in standard English?

Reviewer #1: Yes

Reviewer #2: Yes

5. Review Comments to the Author

Reviewer #1: Message to the Authors:

This review is for the manuscript, “Towards women-inclusive ecology: Representation, behavior, and perception of women at an international conference” (PONE-D-21-16326). The authors investigated the attendance and participation (as both speakers and audience members) of women at the 1st Meeting of the Iberian Society of Ecology, as well as potential differences in perception of inclusion and community between female and male participants. Through documentation of the invited speaker panels, the authorship of research presentations, and the gender of those asking questions following research presentations, as well as a post-meeting survey, the authors documented a relative lack of female last authors (signifying a lack of female-led research groups presenting) as well as a lack of female invited keynote speakers. Women also appeared to ask fewer questions and, from the post-meeting survey, feel less included and free to ask questions than their male peers. Results varied based on professional demographics (e.g., early career vs. permanently positioned later career participants), with those women who had “stuck it out” responding in surveys that they felt free to ask questions and felt professionally satisfied and included at the meeting. These results suggest that professional meetings should evaluate methods for inviting speakers and including the entire community in question and answer/discussion times to better accommodate all attendees. Although PLoS ONE does not evaluate manuscripts based on their novelty or marketability, this manuscript contributes toward a better understanding of how women interact within the academic environment and suggests several paths toward better inclusion, which, as the authors note, is critical to preventing professional stagnation and loss of women from the academic ecological workforce. Analyses were appropriately performed, and the manuscript is technically sound. I have a few concerns regarding the conclusions drawn from the data, but with revisions to better clarify what conclusions can be drawn from the self-selected survey as well as how global the application of these results may be, given the focus of the conference being the Iberian Peninsula, I believe this research is scientifically sound and makes a meaningful contribution to our knowledge.

Attached are the comments on major and minor issues I would recommend be addressed. I want to be very clear that none of the below issues reflect my personal opinion on or experience with gender discrimination, but are questions and concerns that occurred to me regarding the data and conclusions presented.

Major Issues

• 32% of attendees completed the survey, and 60% of those were women, which is significantly different from the composition of attendees. Further discussion of the potential influence of self-selection in responding to the survey is warranted. The authors question the self-reporting of women in the discussion, further calling into question the survey results. This needs to be addressed.

• I am concerned about the temporal aspect of the conclusions regarding last authorship. If it takes a significant amount of time to rise to the level of PI that would produce a large quantity of research, how can we conclude that the lack of female last authorship at this conference isn’t a product of conditions that discouraged female participation at upper-level research institutions a decade ago, rather than some factor of invitation or current environment? The same concerns would arise for conference organizers and committee leaders. These positions require time to achieve, so I do wonder whether this is a bias or a lack of women at high-level (i.e., R1) institutions that have the funding and time to guide, support and participate in these roles, and whether the early-career female attendees signal any movement away from this situation. I would like to see some discussion of this possibility.

• 83% of respondents were of Spanish nationality, and most were living in Spain during the time of the conference. While the conference was international, I do wonder how cultural differences may play a role in the global applicability of the results. While I do not suspect that the overall conclusions and recommendations should be altered, I think this is a point worth discussing, particularly regarding the cultural perceptions of women in the workforce, domestic obligations, parenthood, familial caregiving, and other factors that vary heavily across cultures.

• The results in Table 2 show statistically significant differences in feeling imposter syndrome and hearing gender stereotypical remarks. While these differences are statistically significant, are the differences significant in the real world, or perhaps an artifact of how survey respondents perceive frequency, potentially based on sensitivity to an event? The numbers for imposter syndrome (Q5) for both women and men would fall in “fairly frequent” or “middle of the road”, while gender stereotypical remarks (Q11) would fall on the “next-to-never” end of the spectrum for both men and women. Are either of these factors, then, truly underlying differences in conference participation?

• Please clarify what specific aspects of Q&A sessions favor men. This topic is mentioned several times in the discussion but is never stated outright in terms of what the authors mean.

Minor Issues

• Lines 89-91: should be “This theory states…”; this description of the theory is also unclear and should be clarified.

• Lines 173-174: this is vague.

• Line 184: should be “latter”

• Lines 340-341: how many more comments, on average?

• Lines 348-349: “bias” can imply intentional unfairness. Unless there is evidence of this, I would consider rewording.

• Line 388: remove “to”

• Lines 391-392: is there some evidence that we can educate our way out of this kind of subtle inequitable situation? Recommendations are given for how to alter conference format, but this suggests that there may be a place for formal education. It seems a bit open here for additional comment/discussion.

• Lines 398-399: the results do not suggest that any action be taken. Is there any evidence that we shouldn’t teach women scientists to assert themselves more aggressively during Q&A? What specific structures and dynamics currently favor men? If the end goal is to increase women’s participation, what has been shown to work? This sentence either needs to be elaborated on with citations or re-worded.

• Line 403: remove “to”

• Line 403-405: Line 406 says that this sentence is speculative. Once you question this result, many other possibilities regarding self-selection/self-reporting within your survey crop up. Perhaps remove speculation entirely or devote a paragraph to exploring the possibilities of this self-selection/self-reporting. It’s likely that men self-report differently, too.

• Lines 424-425: Among those who responded to the survey.

• Line 427-430: should be “self-report”; also, there is the possibility that women who encountered sexist attitudes were more likely to respond to the survey.

• Line 435: Please define earlier what specific dynamics of a Q&A are favorable to men. Also, is the exclusion of women from social events direct or indirect?

• Line 462: what kind of clear guidelines and criteria? Are there any that have shown promise, such as a quota system, or are there other frameworks?

• Line 480: should this be “to ask” rather than “to think”? Also, while I think turn taking has promise, forcing attendees to participate seems aggressive to me, in that some attendees may prefer to just listen and learn at their first conference or at a conference on a subject they are just beginning to explore.

• Lines 489-491: citations needed

Reviewer #2: General summary

The paper analyses the representation, behavior and perception of woman at the international conference of the 1st Meeting of the Iberian Society of Ecology. The authors used a novel, multidimensional approach to analyze women’s representation, behavior and perception at a medium sized conference as their study object with a good participation (766 participants at the conference), 56% of the 375 talks analyzed for women’s behavior and 32% of all attendees attending the post-conference survey on their perception.

My impression of the manuscript is that the available data is analyzed thoroughly and that the conclusions from the data are applicable. The title reflects the contents of the paper well, without overselling the results. The abstract shows the results and their interpretation in a too simplified way and in my opinion overstates some of the facts (please see my comments under detailed examination) which could lead to a misconception for the reader and should be, in parts, rewritten more carefully. The introduction is concise and gives a good overview over the topic while leading up to the aims of the study. The results are concise and well put together to address the raised questions and the discussion is to the point while giving good explanations of the results. However, in the introduction it was mentioned that the multidimensional approach should be used to detect mechanisms behind the gender bias. I do not think that with the approach used here mechanisms can be detected, thus the explanations given in the discussion are hypothetical (which is correctly stated in the discussion). I think the authors should be clearer here, their analysis shows nicely women’s representation, the audience’s behavior and its perception of this behavior, however, there was no question ask during the study as to the ‘why’ (mechanisms) of certain behavior. I do think however, that the study has merit even, when hypothesizing on these mechanisms, but this should be made clear.

Detailed examination

Abstract

• Line 51: It is true that women might experience sexism at conferences, however, in your case, you stated there was none overtly sexism reported. I find that statement in the first sentence of the abstract misleading for the expectations on the whole paper. I would rather argue, that conferences are showing a cross section of the scientific community in a given field and are thus ideal study objects for the analysis of gender imbalances, as you have conducted.

• Line 59: The statements on the possibility of women to ask questions and their intellectual satisfaction are overselling the results in my opinion; i.e. the statement ‘most women were unable to ask the questions they wanted’ implies someone or something prevented them actively from asking a question. While as reported in line 400 (self-report) it says ‘women self-reported to have asked questions whenever they wanted less frequently’ does not imply any active bias against women. Also, although the difference between men and women in their frequency to answer questions whenever they wanted in Fig 5. is significant, it also shows that most men did not always asked questions whenever they wanted either, with only a difference in career stage. This should not be oversimplified and formulated more carefully. The authors stated quite correctly in the discussion, that there are various reasons, e.g. a short time frame for the Q&A sessions, which let to women not asking questions and they also stated that their interpretation of the behavior is speculative. Thus, I think the formulation in the abstract should be more oriented on the actual question which was answered by the participants in the post-conference survey. The same goes for: ‘most women […] showed limited intellectual satisfaction’: in table 1 Q7 of the post-conference survey, woman had a satisfaction value of 4.06 ± 0.80 on a scale of 1-5 while men had a value of 4.16 ± 0.78 and the difference between men and women was not significant. Thus, I think the statement, that ‘the post-conference survey revealed that most women were unable to ask the questions they wanted and showed limited intellectual satisfaction during the conference’ is overselling the results and misleading for the reader and should thus be carefully rewritten.

Introduction:

• Line 113: I do not think that with the studied angles -representation, behavior and perception - can detect mechanisms behind gender bias. I rather think that using a multidimensional approach gives a more comprehensive picture, but to detect mechanisms, questions about the ‘how’ of a certain behavior would need to be addressed.

Methods

• Line 126: (max. one per session)

• Line 157: true for 97% of cases – how was this tested? Was this part of the survey? Or based on who was first author, versus who presented the study? Did the presenter not need to be first author?

• Line 178: mismatch observed with [noted?] gender – Could this create a problem in the analysis?

• 183/184: ‘… while the most frequent answer was computed for gender-related questions.’ I do not understand this, please rephrase.

• Also on line 183: A praise for the study design: if the inter-observer variability was <5%, the protocol was very well thought out.

• Line 192: Why were the sessions not independent? Because the audience can be the same between sessions?

Discussion

• Line 360: general sessions were mostly convened by men: Could this be a historical artifact? General sessions are mostly the same each year, thus it could be likely that the conveners are the same each time?

• Line 377: In the sessions where there were mostly women as conveners there were also more female keynotes: this could also show networks. Conveners are likely to ask people from their own networks to give a keynote speech (know the persons talking style). Thus, if men have a network dominated by men but woman have more other women on their network this would explain this result.

• Line 383: could this bias also be explained by the invited female speakers being less well known as the male ones? (e.g. check citations or maybe also age of the presenters) See also Maas et al 2021, DOI: 10.1111/conl.12797, on women being underrepresented in the list of top-publishing authors, thus they are less visible, and thus less desirable for keynote positions?

• Lines 389-392: if women and men are both influenced by the mentioned implicit bias, how can these results agree with the idea that women are less capable of doing science than men? Should a female scientist not have a higher opinion of the capability of other women, since she is a woman and a scientist herself? The argument has merit when discussion why less women choose science as a career path than men, but I don’t think it explains the lesser attendance at talks given by female scientists if the participation of men and women in these talks was equal.

• Line 427: self-report

• Line 435: please give an example how women are excluded from social events during this conference

6. PLOS authors have the option to publish the peer review history of their article (what does this mean?). If published, this will include your full peer review and any attached files.

Reviewer #1: No

Reviewer #2: **Yes: **Dr. Sandra Hille

---

## [Author Response · Author response to Decision Letter 0]

8 Sep 2021

Dear editor and reviewers of PLOS ONE, 

Please find enclosed our responses to the reviewers’ comments regarding the paper “Towards women-inclusive ecology: Representation, behavior, and perception of women at an international conference” (PONE-D-21-16326). We are glad that the reviewers found the study significantly sound, and their criticisms and suggestions have been of great help in improving the manuscript.

In this revision, we took into consideration the comments raised by the reviewers and worked thoughtfully to address them all. Following the reviewers’ suggestion, we have rewritten parts of the discussion to clarify our findings and their potential causes. Specifically, we explain the results of the post-conference survey in more detail, provide a detailed description of the demographic inertia, clarify how women are excluded from Q&A dynamics and social events, and give specific examples on how to generate inclusive conferences. Also, we now clearly state the methodological constrains of our study and justify their global applicability. Finally, we have rewritten the abstract and introduction to avoid misconceptions.

Below you will find detailed responses to each of the comments. For each comment, we specify the changes made in the manuscript and their location. In case where we did not make requested changes, we give a detailed explanation stating why. Please, do not hesitate to contact us if further clarifications are needed.

Sincerely,

Anna Lupon (on behalf of all coauthors)

---

## [Decision Letter · Decision Letter 1]

4 Nov 2021

Towards women-inclusive ecology: Representation, behavior, and perception of women at an international conference

PONE-D-21-16326R1

Dear Dr. Lupon,

We’re pleased to inform you that your manuscript has been judged scientifically suitable for publication and will be formally accepted for publication once it meets all outstanding technical requirements.

Kind regards,

Ute Risse-Buhl, Ph.D.

Academic Editor

PLOS ONE

Additional Editor Comments (optional):

Reviewers' comments:

Reviewer's Responses to Questions

**Comments to the Author**

1. If the authors have adequately addressed your comments raised in a previous round of review and you feel that this manuscript is now acceptable for publication, you may indicate that here to bypass the “Comments to the Author” section, enter your conflict of interest statement in the “Confidential to Editor” section, and submit your "Accept" recommendation.

Reviewer #1: All comments have been addressed

Reviewer #2: All comments have been addressed

2. Is the manuscript technically sound, and do the data support the conclusions?

Reviewer #1: Yes

Reviewer #2: (No Response)

3. Has the statistical analysis been performed appropriately and rigorously? 

Reviewer #1: Yes

Reviewer #2: (No Response)

4. Have the authors made all data underlying the findings in their manuscript fully available?

Reviewer #1: Yes

Reviewer #2: (No Response)

5. Is the manuscript presented in an intelligible fashion and written in standard English?

Reviewer #1: Yes

Reviewer #2: (No Response)

6. Review Comments to the Author

Reviewer #1: All concerns were adequately addressed, and this manuscript addresses an important and compelling question with a solid analytical approach. The material you have added makes your case much more compelling. The way you explained the Q&A dynamics and how they could be improved was useful. I have a few comments below just to help with final editing:

Line 95: should be “pursue” not “pursuit”

Line 100: consider “do not belong in…” not “do not belong to…”

Lin 178: “explain” instead of “explained”?

Line 323: “only a few respondents…”

Line 388: “because of higher share of domestic tasks, and parental or elderly care.” Is either grammatically incorrect or just an awkward phrase. Needs to be re-phrased somehow. I agree with the thought, though.

Line 398: should be “young women’s motivation”

Line 419: “urging” rather than “bursting”?

Line 524: remove “out”

Reviewer #2: (No Response)

7. PLOS authors have the option to publish the peer review history of their article (what does this mean?). If published, this will include your full peer review and any attached files.

Reviewer #1: No

Reviewer #2: **Yes: **Sandra Hille

---

## [Editor Report · Acceptance letter]

3 Dec 2021

PONE-D-21-16326R1 

Towards women-inclusive ecology: Representation, behavior, and perception of women at an international conference 

Dear Dr. Lupon:

I'm pleased to inform you that your manuscript has been deemed suitable for publication in PLOS ONE. Congratulations! Your manuscript is now with our production department. 

Kind regards, 

on behalf of

Dr. Ute Risse-Buhl 

Academic Editor

PLOS ONE